# 🍩 InfBaGel: Human-Object-Scene Interaction Generation with Dynamic Perception and Iterative Refinement

**Yude Zou**[1,2][*], **Junji Gong**[3], **Xing Gao**[2][†], **Zixuan Li**[4], **Tianxing Chen**[5], **Guanjie Zheng**[1]
[1]Shanghai Jiao Tong University,  [2]Shanghai Artificial Intelligence Laboratory,
[3]Sichuan University,  [4]Shenzhen University,  [5]The University of Hong Kong

## Abstract

Human–object–scene interactions (HOSI) generation has broad applications in embodied AI, simulation, and animation. Unlike human–object interaction (HOI) and human–scene interaction (HSI), HOSI generation requires reasoning over dynamic object–scene changes, yet suffers from limited annotated data. To address these issues, we propose a coarse-to-fine instruction-conditioned interaction generation framework that is explicitly aligned with the iterative denoising process of a consistency model. In particular, we adopt a dynamic perception strategy that leverages trajectories from the preceding refinement to update scene context and condition subsequent refinement at each denoising step of consistency model, yielding consistent interactions. To further reduce physical artifacts, we introduce a bump-aware guidance that mitigates collisions and penetrations during sampling without requiring fine-grained scene geometry, enabling real-time generation. To overcome data scarcity, we design a hybrid training strategy that synthesizes pseudo-HOSI samples by injecting voxelized scene occupancy into HOI datasets and jointly trains with high-fidelity HSI data, allowing interaction learning while preserving realistic scene awareness. Extensive experiments demonstrate that our method achieves state-of-the-art performance in both HOSI and HOI generation, and strong generalization to unseen scenes. Project page: yudezou.github.io/InfBaGel-page.

## 1 Introduction

In daily life, human activities frequently involve manipulating objects while navigating cluttered environments and interacting with surrounding static elements. For instance, an individual might carry a chair to a target location, avoid obstacles, place it down, and then sit on it. Generating such integrated interactions, known as human-object-scene interaction (HOSI) generation, is essential to advance animation, simulation, and game applications.

HOSI generation requires reasoning about the complex interplay among the human, the manipulated object, and the surrounding scene, introducing several challenges: (i) *Dynamic Object–Scene Changes*: The scene context evolves continuously with human navigation, while the manipulation of sizable objects further reshapes the scene layout. These dynamic changes complicate the generation of consistent interactions among humans, objects, and scenes elements. (ii) *Annotated Data Scarcity*: Capturing high-quality, diverse human-object-scene motion data is challenging due to the combinatorial explosion of object types, scene configurations, and task instructions, which requires substantial human and computational resources. The resulting lack of diverse datasets with comprehensive scene annotations hinders the development of generalizable generative models.

Recent advances have improved either human-object interaction (HOI) or human-scene interaction (HSI) generation, but they remain insufficient in addressing the multifaceted challenges of HOSI. For example, recent scene-aware pipelines (Jiang et al., 2024b;a) adopt a one-off static scene encoding, generating motion without updating the scene state influenced by human or object movement. Additionally, some stage-wise decomposition methods

---

[*]Work done when Yude Zou was an intern at Shanghai Artificial Intelligence Laboratory.
[†]Corresponding Author.

(Zhang et al., 2024; Yao et al., 2025) separate locomotion from interaction, compromising temporal consistency. Planner-dependent approaches (Yi et al., 2024; Geng et al., 2025) introduce computational costs and depend heavily on planner quality. At the data level, existing datasets remain limited. Some (Li et al., 2023) lack scene annotation, while others (Jiang et al., 2024b;a) offer restricted object diversity and lack diverse large manipulable objects. These gaps motivate a scene-aware and data-efficient HOSI generation framework.

To address these challenges, we propose INFBAGEL, an autoregressive framework that generates HOSI from high-level instructions such as goal locations and textual commands. Our approach adopts a coarse-to-fine generation scheme with a consistency model that performs few-step iterative refinement during denoising, conditioned on dynamic perceptual representations of the environment, achieving substantially faster inference than diffusion-based baselines. Furthermore, we employ a hybrid data training strategy that eliminates the need for annotated HOSI data, enabling the model to learn diverse, complex, and coherent interactions. Although trained for HOSI, the model generalizes naturally to various HSI tasks, like locomotion and static-object interactions. The INFBAGEL (generating interaction from bump-aware guidance in real-time) is characterized by:

- A coarse-to-fine conditional interaction generation framework aligned with the few-step denoising of a consistency model, in which a dynamic perception strategy adaptively updates scene context and a bump-aware guidance further improves physical plausibility.
- A hybrid data training strategy that combines synthetic HOSI data with high-fidelity HSI data, relaxing the reliance on complete HOSI annotations and addressing data scarcity.

Specifically, for the challenge of dynamic object–scene, INFBAGEL first generates a coarse motion sequence conditioned on the initial scene. This initial motion is then used to derive time-varying scene state, which in turn guide an iterative optimization process with bump-aware guidance. We begin by training a scene-conditioned diffusion model that accepts the temporal-aligned scene states as a condition and supports conditional and unconditional generation, empowering unified generation for coarse trajectory and fine motion. We further distill the diffusion model into a consistency model, whose few-step high-quality motion generation allows us to update the precise time-varying scene more reliably, and then refine the motion by the latest scene states. Furthermore, we introduce a bump-aware guidance in sampling that guides the model toward collision-free optimization.

To tackle the scarcity of annotated data, we propose a hybrid data training strategy that jointly leverages HSI datasets with explicit scene geometry and HOI datasets with synthesized voxelized scene context. For HOI data, we first identify the volume occupied by the human and the object, and then fill the surrounding free space with voxels, transforming the original HOI data into human–object–scene tuples without manual scene annotation or high-fidelity mesh capture. Joint training on this mixture enables the model to disentangle human, object, and scene factors, achieving strong zero-shot scene generalization for HOSI generation. This hybrid strategy broadens data scale and diversity, supplies consistent spatial context for collision reasoning, and provides a practical, data-efficient training foundation. Extensive experiments demonstrated that INFBAGEL achieves the highest success rate and physical feasibility in HOSI generation, while maintaining highly competitive performance on the HOI benchmark.

Beyond such technical contributions, INFBAGEL is promising for various downstream applications, such as robot learning and virtual character animation. For example, in humanoid motion and robot learning, INFBAGEL acts as a text-conditioned high-level motion planner that captures rich human-object-scene interaction patterns, while low-level controllers handle physics-based tracking and embodiment-specific constraints. Its strong zero-shot generalization, validated on the HOSI benchmark with 67 unseen scenes, demonstrates its adaptability and effectiveness in diverse real-world scenarios.

## 2 RELATED WORK

**Interaction with Objects.** Regarding **interaction with small objects**, recent research has shifted from synthesizing hand motions (Zhang et al., 2021; Christen et al., 2022) to

full-body motions for object grasping (Taheri et al., 2022; Wu et al., 2022). Some methods simultaneously generate human motion and predict object motion (Braun et al., 2024; Ghosh et al., 2023); however, these approaches are mainly limited to interactions with small objects, with an emphasis on hand motion. For **interaction with large objects**, the complexity of human motion increases, and object motion becomes non-negligible. Some methods employ reinforcement learning strategies to synthesize actions for specific tasks, such as box moving (Merel et al., 2020). Recent mimic learning methods (Xu et al., 2025; Yu et al., 2025) achieve generalized character interaction. With the growing availability of human-object interaction datasets (Bhatnagar et al., 2022; Li et al., 2023), certain methods (Li et al., 2024b; Cong et al., 2025) start generating motions for interactions with large objects, often relying on sequential points or object trajectories, which restricts the model's ability to autonomously generate diverse interactions. Other methods (Xu et al., 2023; Peng et al., 2025; Li et al., 2025; Xue et al., 2025) attempt to simultaneously synthesize human and object motions using multi-stage strategies.

**Interaction with Scene.** Early work primarily focuses on generating **interactions within static scenes**, extending from locomotion in the scene (Zhang & Tang, 2022; Mao et al., 2022) to interacting with static objects (Zhang et al., 2020b; Wang et al., 2022b). Subsequently, some methods consider both locomotion and static interaction simultaneously. Hassan et al. (2021), Wang et al. (2022a), and Huang et al. (2023) modeled them separately, resulting in inconsistent motions. Zhao et al. (2023) and Yi et al. (2024) rely on high-level path planners, making sensitivity to the quality of the planner. Wang et al. (2024b) and Cen et al. (2024) are multi-stage frameworks, which incur significant computational overhead. Recently, generating **interaction with dynamic objects** in the scene gains increasing attention. Some physics-based methods (Hassan et al., 2023; Xu et al., 2024; Pan et al., 2025; Wang et al., 2024a) explore the interactions between physically simulated humanoids and dynamic objects through reinforcement learning. However, these methods often require complex reward engineering and still struggle with motion diversity and realism. Kinematics-based methods Jiang et al. (2024b) and Jiang et al. (2024a) propose a unified generation framework, but simplify object motion by directly attaching objects to the hand. Yao et al. (2025) and Geng et al. (2025) further generate object motion. However, Yao et al. (2025) is limited in motion and object types due to the limited HOSI dataset. Geng et al. (2025) synthesizes HOSI data by matching OMOMO motions to collision-free scenes, but unmatched cases are simply treated as empty scenes. Both approaches rely on static scene perception and additional path planners, limiting their ability to handle dynamic contexts. In this paper, we focus on kinematics-based generation and propose INFBAGEL, which accommodates diverse motion types and scenes through a hybrid data training strategy. It is designed as a unified coarse-to-fine framework with few-step consistency sampling and dynamic perception, enabling more adaptive and realistic HOSI interactions.

**Interaction Datasets.** Current human-object interaction datasets (Li et al., 2023; Bhatnagar et al., 2022) generally lack scene context, which makes some methods only integrate the generated motion into the external scene by planning collision-free paths at inference time. However, these methods lack direct penetration awareness, limiting their applicability in complex 3D environments. Alternatively, Liu et al. (2024) aggregate motion-only data into paired human-occupancy interaction data to synthesize HOSI data. Rather than a trivial extension, our approach further incorporates real high-fidelity HSI data through a hybrid training strategy, effectively learning HOSI while ensuring realistic scene awareness. Recently, some work introduces object interactions into HSI datasets. For example, Jiang et al. (2024b) includes object motion but lacks instruction-level annotations, and the variety of objects is restricted. Jiang et al. (2024a) features diverse scenes and text annotations but omits object motion. Kim et al. (2025) has a limited variety of scenes, making it difficult to achieve generalized scene perception. The restricted variety of objects and scenes limits the learning of complex interactions in generation. More discussions provided in Appendix A.4

## 3 METHODOLOGY

We introduce INFBAGEL, an auto-regressive framework for generating human-object-scene interactions from textual instructions and goal locations, as illustrated in Fig. 1.

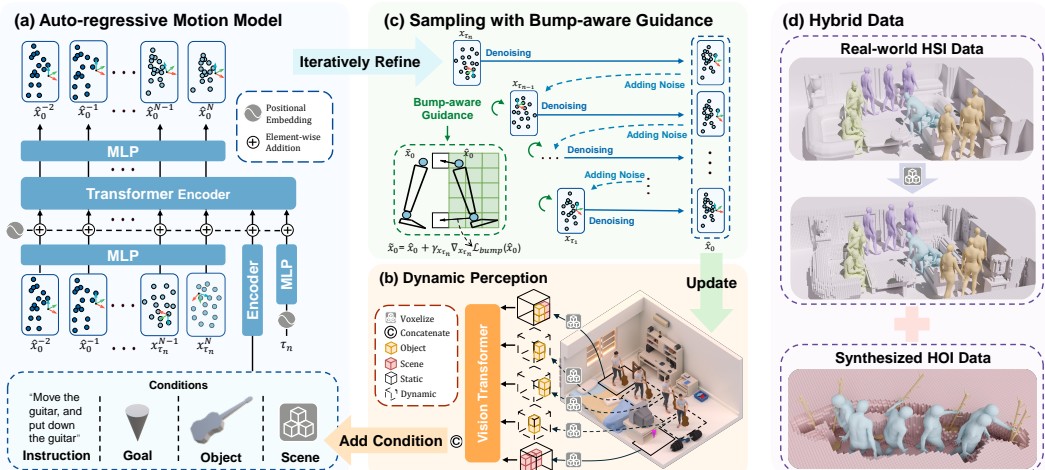

Figure 1: Overview of INFBAGEL. Our method operates through an iterative refinement process. (a) Auto-regressive Motion Model generating arbitrary long-sequence motions conditioned on textual instructions, goals, object geometry, and scene context. (b) Dynamic Perception Encoder perceives the evolving environment with the temporal-aligned scene state updated by iterative sampling. (c) Bump-aware Guidance detects collisions and directs iterative, collision-free sampling. (d) Hybrid Data training enables robust zero-shot generalization to complex realistic scenes.

## 3.1 DATA REPRESENTATION

**Problem Formulation.** Formally, our objective is to generate a coupled motion sequence $\mathcal{M} = (\mathcal{M}_h, \mathcal{M}_o)$ including both human motion $\mathcal{M}_h$ and object motion $\mathcal{M}_o$, given the 3D scene $\mathcal{S}$, the interactive object geometry $\mathcal{O}$, the text instruction $\mathcal{T}$, and the goal position $\mathcal{G}$. For the interaction with static scene, the object motion $\mathcal{M}_o$ will be set to empty.

**Human and Object Motion.** At each frame $t$, the human motion $\mathcal{M}_h$ consists of the root translation $\mathbf{T}_h \in \mathbb{R}^3$ and 6D rotation representations $\mathbf{R}_h \in \mathbb{R}^{J \times 6}$ for $J = 22$ selected joints. We use the parameterized human model SMPL-X (Pavlakos et al., 2019) to reconstruct the human mesh. The object motion $\mathcal{M}_o$ is represented by the translation of its centroid $\mathbf{T}_o \in \mathbb{R}^3$ and the rotation $\mathbf{R}_o \in \mathbb{R}^{3 \times 3}$ relative to the input object geometry $\mathcal{O}$.

**Input Conditions.** We represent the interactive object geometry $\mathcal{O}$ using Basis Point Set (BPS) representation (Prokudin et al., 2019), which consists of 1024 vectors indicating the spatial relationship from a set of basis points to the object's surface, denoted as $\mathcal{O} \in \mathbb{R}^{1024 \times 3}$. The 3D scene $\mathcal{S}$ is represented with voxel grids. The goal location is a user-specified 3D coordinate $\mathcal{G} \in \mathbb{R}^3$, including human pelvis goal and object centroid goal. For static scene interaction tasks, the geometry and goal of interactive object is set to empty. The text instruction $\mathcal{T}$ is encoded into a 768-dimensional feature using CLIP (Radford et al., 2021).

## 3.2 DYNAMIC PERCEPTION ENCODER

Building on the above objective, the traversable space evolves as the human and object move, so a one-off static encoding is insufficient for HOSI generation. To address this challenge, we introduce a coarse-to-fine generation strategy with dynamic perception encoding mechanism as showed in Fig. 1(b). INFBAGEL can generate a coarse motion trajectory to derive frame-wise, time-varying scene states that capture how the environment changes over time, and then optimize the motion based on updated scene state correspondingly.

Following Jiang et al. (2024b) and Jiang et al. (2024a), we represent scene states by 3D occupancy voxel. Within each generation window, we use five scene voxel grids to represent local scene information, three dynamic and two static. Specifically, the two static voxel grids respectively capture the start and goal regions, while the three dynamic voxels centered at the pelvis positions uniformly sampled from intermediate frames within the temporal window. The dynamic voxels are masked out during coarse prediction, and updated iteratively as the

prediction is refined. Each voxel grid is a 3D array $\{0, 1, 2\}^{N \times N \times N}$, where 0, 1, and 2 denote traversable, non-traversable, and interactive object occupied cells, respectively, and $N$ is the grid size. We then employ a ViT (Dosovitskiy et al., 2020) to independently encode each scene voxel grid to a 512-dimensional embedding. To incorporate precise spatial cues, we further fuse the positional encoding of the central voxels, *i.e.,* the corresponding pelvis/goal positions, with respective scene embeddings. More details are provided in Appendix A.6.

## 3.3 Auto-regressive Motion Model

Regarding the coarse prediction, existing methods usually introduce additional modules or algorithms, which is cost for long horizons. To achieve unified generation, we exploit the iterative generation nature of the diffusion model to perform coarse-to-fine generation, where the dynamic scene conditions are updated based on the results from the previous iteration, and each iteration further refines the generation according to the updated scene state. Specifically, we train a diffusion model conditioned on a combination of scene, text, goal and object information, with a random mask applied to the embeddings of dynamic voxel grids. This mechanism enables the model to seamlessly handle both static and dynamic scene perceptions, generating coarse trajectories without dynamic embeddings and producing fine-grained motion refinements when dynamic cues are available.

However, the diffusion model typically requires a large number of sampling steps to generate a clean sample, which compromises generation efficiency and hinders the accurate update of scene states. To address this limitation and construct a more efficient framework, we distill the diffusion model into a **consistency model**, which can generate high-quality motion sequences within only a few steps. Notably, each step in the consistency model directly outputs a clean sequence, unlike the noisy intermediate states of diffusion models, thereby providing a more reliable perception of the evolving scene for subsequent refinement.

Specifically, the consistency model $f_\theta$ serves as a distilled version of the diffusion model, learning to directly map any noisy sample $\mathbf{x}_{\tau_n}$ along a denoising trajectory back to its clean origin $\hat{\mathbf{x}}_0$, with $\tau_n$ representing the diffusion step. Under the classifier-free guidance setting, the consistency model $f_\theta : (\mathbf{x}_{\tau_n}, \tau_n, \mathbf{C}_{\tau_n}, \omega) \mapsto \hat{\mathbf{x}}_0$ jointly models the conditional and unconditional settings, where the interpolation weight $\omega$ controls the guidance strength and $\mathbf{C}_{\tau_n}$ denotes the conditions including scene, textual instruction, goal, and object information. To effectively transfer the knowledge of multi-step diffusion model into $f_\theta$, we employ **consistency distillation (CD)**, which enforces local consistency between the outputs of $f_\theta$ at adjacent PF-ODE steps. Mathematically, following Luo et al. (2023), the distillation objective minimizes

$$\mathcal{L}_{\text{CD}} = \mathbb{E}_{\mathbf{x},n,\mathbf{C},w}\Big[d\big(f_\theta(\mathbf{x}_{\tau_n}, \tau_n, \mathbf{C}_{\tau_n}, \omega), f_{\theta'}(\hat{\mathbf{x}}_{\tau_{n-1}}^{\Psi,\omega}, \tau_{n-1}, \mathbf{C}_{\tau_{n-1}}, \omega))\Big], \tag{1}$$

where the parameters $\theta'$ of the target network $f_{\theta'}$ are updated with the exponential moving average of the trainable parameters $\theta$ of the online network $f_\theta$, and $d(\cdot, \cdot)$ is the $\ell_2$ distance. Here, $\hat{\mathbf{x}}_{\tau_{n-1}}^{\Psi,\omega}$ is estimated from $\mathbf{x}_{\tau_n}$ by the teacher diffusion model using DDIM (Song et al., 2020)) sampling ($\Psi$) under classifier-free guidance with strength $\omega \in [\omega_{min}, \omega_{max}]$:

$$\hat{\mathbf{x}}_{\tau_{n-1}}^{\Psi,\omega} = \mathbf{x}_{\tau_n} + (1 + \omega)\Psi(\mathbf{x}_{\tau_n}, \tau_n, \mathbf{C}_{\tau_n}) - \omega\Psi(\mathbf{x}_{\tau_n}, \tau_n, \mathbf{C}_{\tau_n}^\emptyset), \tag{2}$$

where $\mathbf{C}_{\tau_n}$ and $\mathbf{C}_{\tau_n}^\emptyset$ represent the full condition and without the dynamic scene condition, respectively.

To further enhance the physical realism of the generated motion, inspired by Li et al. (2024b), we incorporate auxiliary supervision on human joints and object poses. Using forward kinematics (FK) to the ground-true motion $\mathcal{M}_{h/o}$ and the predicted motion $\hat{\mathcal{M}}_{h/o}$, we compute the positions of hands and feet, as well as the transformed object vertices. The loss is then calculated as follows:

$$\mathcal{L}_{\text{joints}} = \big\|FK_h(\mathcal{M}_h) - FK_h(\hat{\mathcal{M}}_h)\big\|_2^2, \tag{3}$$

$$\mathcal{L}_{\text{obj}} = \big\|FK_o(\mathcal{M}_o) - FK_o(\hat{\mathcal{M}}_o)\big\|^2. \tag{4}$$

The total objective is computed as a weighted sum of such loss functions, with hyperparameter $\lambda_h$, $\lambda_o$ as loss coefficients

$$\mathcal{L} = \mathcal{L}_{\text{CD}} + \lambda_h \mathcal{L}_{\text{joints}} + \lambda_o \mathcal{L}_{\text{obj}}. \tag{5}$$

During inference, we leverage the consistency model to directly generate reliable motion. To synthesize long-horizon interactions, as illustrated in Fig. 1(a), we utilize the efficiency of the consistency model by adopting an auto-regressive generation strategy. Rather than generating the entire sequence at once, the model produces motion segments sequentially. This design improves the efficiency and effectiveness of updating the scene state, making real-time iteration feasible.

### 3.4 BUMP-AWARE GUIDANCE

Despite dynamic-perception iterations, strict collision avoidance remains challenging. To steer refinement toward collision-free solutions while preserving the efficiency of consistency-based optimization, we introduce a lightweight bump-aware guidance (Fig. 1c).

Specifically, at each sampling iteration, the consistency model predicts the clean data $\hat{\mathbf{x}}_0$ which derive $\hat{\mathcal{M}}_h$ and $\hat{\mathcal{M}}_o$ to reconstruct the human joints and object points. Given the voxelized scene $\mathcal{S}$, we detect bumps when overlaps occur between human or object points and occupied voxels, and use the distance with closest free space to calculate gradient with respect to the current motion to nudge samples away from obstacles. Following Dhariwal & Nichol (2021) and Ho et al. (2022), the process is formulated as:

$$\tilde{\mathbf{x}}_0 = \hat{\mathbf{x}}_0 + \gamma_{\tau_n} \nabla_{\mathbf{x}_{\tau_n}} \mathcal{L}_{\text{bump}}(\hat{\mathbf{x}}_0), \tag{6}$$

$$\mathcal{L}_{\text{bump}} = \sum_{p \in \{\hat{\mathcal{M}}_h, \hat{\mathcal{M}}_o\}} D(V(p)) \tag{7}$$

where $\gamma_{\tau_n}$ is the guidance scale related with $\tau_n$, $V(\cdot)$ denotes the voxel occupied by the point. $D(\cdot)$ returns the distance from the voxel center to the closest free voxel center, which is 0 if the voxel itself is free.

Bump-aware guidance does not rely on high-fidelity meshes. The regular structure of voxel grids facilitates the pre-computation of a distance map, thereby avoiding the costly online nearest point search required by detailed meshes and reducing computational overhead. It integrates seamlessly into the consistency sampling loop, progressively reducing penetration while enhancing overall physical plausibility.

### 3.5 HYBRID DATA TRAINING

Another major bottleneck for HOSI generation is the scarcity of datasets that simultaneously offer diverse scenes, a wide variety of manipulable objects, and textual annotations, which limits generalization if trained on a single source. To broaden coverage while preserving the unified interface $(\mathcal{S}, \mathcal{O}, \mathcal{T}, \mathcal{G})$, we introduce a hybrid data training strategy that jointly combines existing high-quality HSI and HOI datasets, as shown in Fig. 1(d). This approach allows us to train a model that is proficient in both complex scene navigation and interaction with static or dynamic object, without requiring a single, all-encompassing HOSI dataset.

To learn diverse and complex scene-level interaction, we leverage HSI datasets such as LINGO (Jiang et al., 2024a). These datasets provide high-fidelity scene meshes, diverse environments, and detailed textual instructions for tasks like locomotion and interaction with static objects (e.g., sitting on a chair). Training on this data enables our model to ground its understanding of motion in realistic, complex geometric contexts and to learn long-horizon planning and navigation behaviors.

To master diverse dynamic object interactions, we incorporate large-scale HOI dataset (Li et al., 2023) with LINGO. Since HOI dataset typically lack scene information, we synthesize a minimal yet effective scene context for each sample. Following the methodology of Liu et al. (2024), we identify the spatial volume occupied by the human and the manipulated object throughout the motion. The surrounding free space are then voxelized to create a plausible

occupied space. This process transforms standard HOI data into human–object–scene tuples without manual scene annotation, effectively teaching the model to respect the spatial constraints imposed on the human and object motion.

The HSI data provides the "macro-level" understanding of interaction within a static world, while the synthesized HOI data offers the "micro-level" knowledge of object manipulation. By jointly training on this heterogeneous data mixture, our model learns consistent collision reasoning for spatial context and yields strong zero-shot generalization to unseen scenes.

## 4 EXPERIMENTS

### 4.1 EVALUATION SETTINGS

**Dataset and Metrics.** We construct an HOSI benchmark as testing set, comprising 469 sequences, featuring 7 categories of large, manipulable objects from OMOMO (Li et al., 2023) located within 67 diverse and **unseen indoor scenes** from TRUMANS (Jiang et al., 2024b). Specifically, each sequence is constructed by first randomly sampling a unique start-end pair within a scene, and then pairing it with a sampled initial state of human as well as interactive object and textual description from the OMOMO-test set. This approach effectively preserves the diverse motion types inherent in OMOMO and the diversity of scenes from TRUMANS. Further details regarding the HOSI benchmark, such as the feasibility validation of the synthetic sequences, and the composition of training set are provided in Appendix A.5. We assess the generated motion from three perspectives. For task completion, we measure the final distance of the human ($T_h$) and the object ($T_o$) to their respective goals, along with the task success rate ($S_\%$), considered successful if both distances are below 10 cm. For motion and interaction quality, we assess foot sliding (FS), contact percentage ($C_\%$) and human-object penetration ($P_{body}$) from CHOIS (Li et al., 2024b). To evaluate scene awareness, we measure scene penetration for both the human and the object using metrics from DIMOS (Zhao et al., 2023), including the average, max penetration vertices depth sum ($P_{mean/max}$) in any frame during the sequence, measured in meters (m), and the percentage of frames with penetration ($P_{f\%}$). For the unified penetration measure, we calculate $P_{body}$ by summing the total depth of penetration from any frame through vertices, rather than the average value, measured in meters (m). For time cost, we follow (Dai et al., 2024) to report the average inference time per sentence (AITS) and FPS.

**Baselines.** We compare INFBAGEL with two scene-aware baselines, TRUMANS (Jiang et al., 2024b) and LINGO (Jiang et al., 2024a). For a fair comparison, we augment both with object geometry, goal location, and output heads for object-motion prediction. We also adapt TRUMANS to accept text instructions and object goals (it originally relies on hard joint constraints rather than goal conditioning). All models are trained on the OMOMO dataset for evaluation across both HOSI and HOI with the same model. For HOSI, training uses OMOMO enriched with synthesized scenes. To assess our hybrid-data training strategy, we additionally train on a mixture of synthesized OMOMO and the LINGO HSI dataset.

### 4.2 EXPERIMENTAL RESULTS AND ANALYSIS

Table 1 and Table 2 present the quantitative results on the HOSI benchmark. INFBAGEL significantly outperforms TRUMANS and LINGO across nearly all metrics. Notably, INFBAGEL achieves a success rate of 83.16%, a substantial improvement over LINGO's 53.09% and TRUMANS' 1.92%, demonstrating INFBAGEL's superior ability to generate goal-directed interactions in complex scenes. In terms of physical plausibility, INFBAGEL achieves the lowest foot sliding and the highest contact percentage with the lowest penetration between the human and object, indicating more realistic human motion and interaction quality. Furthermore, INFBAGEL exhibits remarkable scene awareness, with human-scene and object-scene penetration metrics significantly lower than those of the baselines, highlighting its effectiveness in avoiding collisions within the environment. LINGO achieves a slightly lower human goal distance ($T_h$), which reflects a deliberate trade-off where our model prioritizes collision-free motion over strict goal adherence presented in the ablation studies. As shown in Fig. 2, our INFBAGEL generates motions and interactions that are

Table 1: Quantitative results of HOSI task, where the human moves the object from one place to another in a cluttered realistic scene. Blue highlights the improvement brought by hybrid data.

| Method | Task Accuracy | | | FS↓ | HO Interaction | | HS Penetration↓ | | | OS Penetration↓ | | |
|---|---|---|---|---|---|---|---|---|---|---|---|---|
| | $T_h \downarrow$ | $T_o \downarrow$ | $S_\% \uparrow$ | | $C_\% \uparrow$ | $P_{body} \downarrow$ | $P_{mean}$ | $P_{max}$ | $P_{f\%}$ | $P_{mean}$ | $P_{max}$ | $P_{f\%}$ |
| *Trained on Synthesized OMOMO Dataset* | | | | | | | | | | | | |
| TRUMANS | - | 71.15 | 1.92 | 0.71 | 46.19 | 4.91 | 10.02 | 70.49 | 93.81 | 56.03 | 218.69 | 34.41 |
| LINGO | **2.75** | 13.03 | 53.09 | 0.57 | 53.83 | 5.72 | 7.62 | 61.19 | 96.15 | 38.07 | 167.95 | 22.93 |
| INFBAGEL | 4.75 | **8.14** | **83.16** | **0.13** | **78.18** | **3.96** | **3.39** | **36.97** | **28.19** | **16.62** | **109.61** | **22.72** |
| *Trained on Hybrid Dataset* | | | | | | | | | | | | |
| LINGO | **3.04** | 17.97 | 39.45 | 0.57 | 53.43 | 7.58 | 7.09 | 57.45 | 91.96 | 22.55 | 123.56 | **20.56** |
| INFBAGEL | 4.37 | **7.94** | **81.45** | **0.15** | **76.96** | **5.05** | **3.17** | **37.09** | **26.74** | **12.45** | **93.37** | 22.32 |

Table 2: Comparison of generation speed. 'DM' denotes Diffusion Model and 'G' denotes Guidance. The average duration of the sequence is about 193 frames.

| Metrics | TRUMANS | LINGO | INFBAGEL $_{DM}$ | INFBAGEL $_{w/o\ G}$ | INFBAGEL |
|---|---|---|---|---|---|
| AITS↓ | 5.84 | 6.46 | 57.17 | **1.30** | 6.75 |
| FPS↑ | 31.57 | 28.86 | 3.38 | **148.54** | 28.75 |

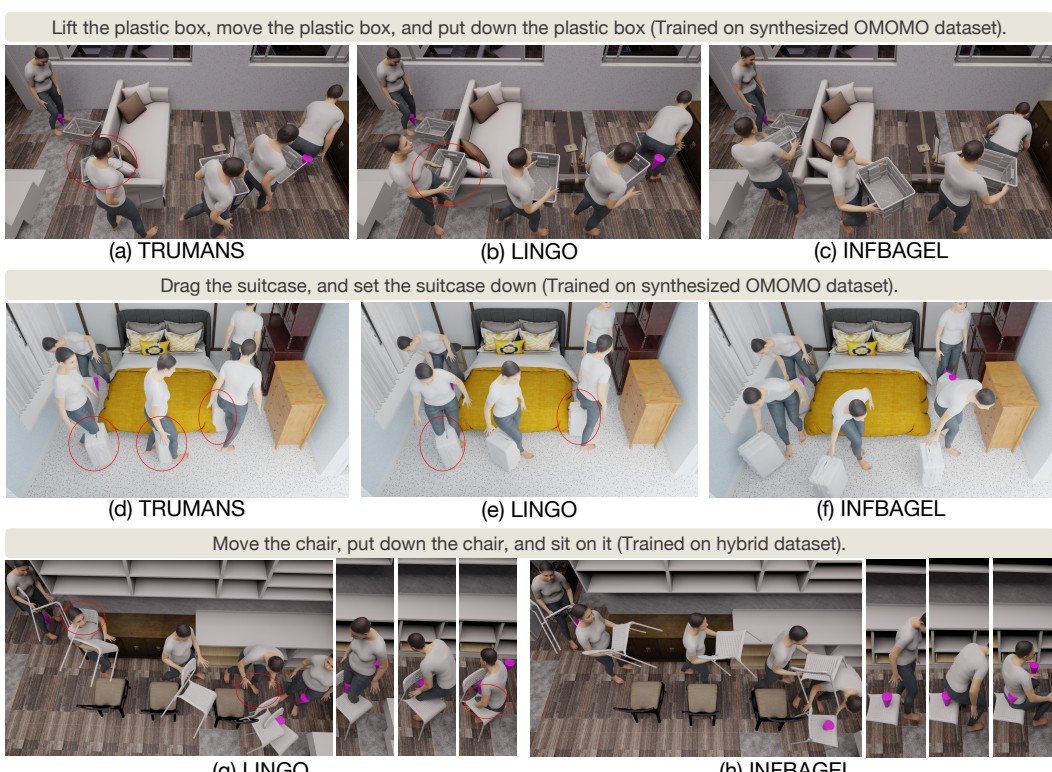

Figure 2: Qualitative comparison. **Top 2 rows:** Comparison on human-object interaction in scenes. **Bottom row:** Comparison on a complex multi-stage task involving moving a chair and then sitting on it.

physically plausible and semantically correct, outperforming baseline methods. Specifically, both TRUMANS (a/d) and LINGO (b/e) exhibit severe object-scene penetration. In contrast, INFBAGEL (c/f) produces nearly collision-free motion. Furthermore, while LINGO (g) fails to generate plausible interactions in both HOI and HSI phases, INFBAGEL (h) successfully executes the long-horizon task by moving the chair to the goal without collision and then seamlessly transitioning to a sitting pose. These results demonstrate the superior physical plausibility and task semantics understanding of our method. To further assess our

Table 3: Ablation on key components: Dynamic Perception Encoding (DP) and Guidance (G). C denotes Contact Guidance (Li et al., 2024b); B denotes our Bump-aware Guidance. A Diffusion Model baseline with DP is included to study effects on dynamic perception, guidance is omitted due to inconsistent sampling steps and weighting.

| Components | | Task Accuracy | | | FS↓ | HO Interaction | | HS Penetration↓ | | | OS Penetration↓ | | | FPS↑ |
|---|---|---|---|---|---|---|---|---|---|---|---|---|---|---|
| DP | G | $T_h \downarrow$ | $T_o \downarrow$ | $S_\% \uparrow$ | | $C_\% \uparrow$ | $P_{body} \downarrow$ | $P_{mean}$ | $P_{max}$ | $P_{f\%}$ | $P_{mean}$ | $P_{max}$ | $P_{f\%}$ | |
| × | × | 2.71 | 7.57 | 71.22 | 0.19 | 65.99 | **3.33** | 6.49 | 56.18 | 35.18 | 28.78 | 146.10 | 23.01 | **180.22** |
| ✓ | × | 2.72 | **6.32** | **86.35** | 0.14 | 68.09 | 4.46 | 6.38 | 55.84 | 29.82 | 26.00 | 138.12 | 23.01 | 148.54 |
| ✓ | C | 2.91 | 6.39 | 85.50 | **0.13** | **79.34** | 3.91 | 6.30 | 57.17 | 28.21 | 25.61 | 139.93 | 23.25 | 29.31 |
| ✓ | C+B | 4.75 | 8.14 | 83.16 | **0.13** | 78.18 | 3.96 | **3.39** | **36.97** | **28.19** | **16.62** | **109.61** | **22.72** | 28.75 |
| Diffusion Model | | **2.12** | 6.63 | 84.22 | 0.14 | 65.14 | 3.96 | 7.20 | 58.09 | 31.14 | 28.48 | 139.69 | 23.82 | 3.38 |

Table 4: Ablation study on different parameter settings in our method, including the models with 1 and 3 dynamic scene representations (Voxel) at different sampling steps (Step).

| Parameters | $S_\% \uparrow$ | FS↓ | HO Interaction | | HS Penetration↓ | | OS Penetration↓ | | AITS↓ | FPS↑ |
|---|---|---|---|---|---|---|---|---|---|---|
| | | | $C_\% \uparrow$ | $P_{body} \downarrow$ | $P_{mean}$ | $P_{max}$ | $P_{mean}$ | $P_{max}$ | | |
| Voxel=1, Step=1 | 68.87 | 0.15 | 66.25 | 4.68 | 6.27 | 58.04 | 27.38 | 139.36 | **0.12** | **1703.79** |
| Voxel=1, Step=8 | 60.98 | 0.16 | 71.29 | 4.27 | 4.07 | 45.52 | 18.64 | 113.30 | 3.11 | 62.35 |
| Voxel=1, Step=16 | 62.26 | 0.15 | 77.32 | 4.20 | 3.81 | 41.87 | 16.74 | 113.31 | 6.57 | 29.53 |
| Voxel=3, Step=1 | **87.85** | 0.14 | 67.62 | 4.78 | 6.15 | 56.61 | 28.16 | 142.22 | **0.12** | 1570.94 |
| Voxel=3, Step=8 | 82.94 | 0.15 | 73.44 | 4.05 | 4.35 | 45.48 | 20.53 | 117.34 | 3.21 | 60.35 |
| Voxel=3, Step=16 | 83.16 | **0.13** | **78.18** | **3.96** | **3.39** | **36.97** | **16.62** | **109.61** | 6.75 | 28.75 |

model's capabilities, we include an additional evaluation on the standard HOI benchmark in the Appendix A.2, where our method still achieves state-of-the-art performance.

**Does hybrid data training strategy help?** As shown in the second group of Table 1, both models trained on hybrid data achieve significantly reduced human and object penetration with scene. This indicates that incorporating HSI data, containing high-fidelity scene geometry, enhances the model's understanding of complex environmental constraints and improves its ability to avoid collisions while navigating the object in scenes. These results demonstrate that our hybrid data strategy contributes to learning generalizable scene representation, enabling the model to perform complex human-object-scene Interaction tasks effectively in unseen environments. Notably, the task type in the LINGO data is different from the HOSI test set, making a slight trade-off in success rate and human-object interaction metrics ($C_\%$ and $P_{body}$), this is expected as the model learns to prioritize scene-level physical plausibility over strictly adhering to interaction priors learned from HOI data. More exploration of the hybrid data ratio is provided in Appendix A.3.

## 4.3 Ablation Study

We conduct a collection of ablation studies to evaluate the contributions of each component on the HOSI benchmark across three dimensions. Specifically, (1) to examine the effect of **dynamic perception encoding (DP)**, we compare a model using only static scene encoding (*i.e.,* without DP) with one adopting DP, as presented in Table 3, and further explore the impact of the number of dynamic voxels in DP, as shown in Table 4. (2) We further study the effect of **bump-aware guidance (B) and contact guidance (C)** on improving physical plausibility, represented as DP+C and DP+C+B in Table 3. (3) Finally, we compare the distilled **consistency model** with a standard diffusion model, both using DP, as shown in Table 3, and study the influence of sampling steps in Table 4. Qualitative comparisons of different variants are provided in Fig. 3.

**Does dynamic perception encoding help?** Comparing the first two rows of Table 3 reveals the impact of our dynamic perception encoding. Disabling this component (row 1) leads to a notable degradation across multiple metrics compared to the model with DP enabled (row 2). In particular, scene penetration by both the human and the object increases, indicating that without dynamically updating its perception of the scene, the model becomes less capable of avoiding collisions and performing fine-grained interaction.

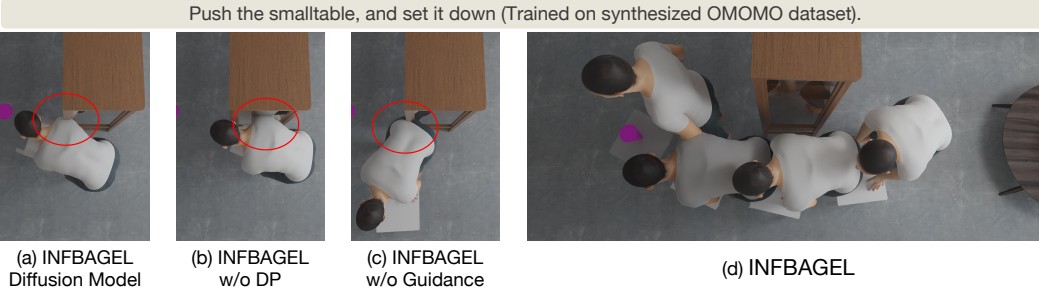

Push the smalltable, and set it down (Trained on synthesized OMOMO dataset).

(a) INFBAGEL Diffusion Model  (b) INFBAGEL w/o DP  (c) INFBAGEL w/o Guidance  (d) INFBAGEL

Figure 3: Qualitative comparison in ablation study. Replacing/removing specific modules: (a) diffusion model instead of consistency model, (b) static perception instead of dynamic perception, and (c) without bump-aware guidance, all resulted in collisions with the scene.

As shown in Fig. 3 (b) and (d), only the static scene encoding results in penetration between the human and scene, while this does not occur when dynamic scene encodings are included. In addition, the overall task success rate drops sharply from 86.35% to 71.22%, and foot sliding worsens from 0.14 to 0.19. This observation is further supported by the results in Table 4. Reducing the temporal voxel number from three to one causes a decline in success rate and worse motion quality in terms of foot sliding. These results underscore that dynamic perception is crucial for generating physically plausible, goal-directed motions in complex and evolving environments.

**Does bump-aware guidance help?** As shown in Table 3 (rows 2–4), introducing contact guidance (C) leads to notable improvements in human-object (HO) interaction metrics, including $C_\%$ and $P_{body}$. Furthermore, incorporating the proposed bump-aware guidance (B) yields consistent and substantial improvement in both HS and OS penetration metrics, while maintaining the HO interaction performance nearly unchanged. Fig 3 (c) and (d) further demonstrate the necessity of bump-aware guidance. These results validate the effectiveness of bump-aware guidance in enhancing the physical plausibility of human-scene and object-scene interactions. However, we also observe that there is a slight degradation in task accuracy metrics such as the goal distance ($T_h$ and $T_o$), which highlights a trade-off introduced by the guidance. Stricter collision constraints enhance physical plausibility by mitigating penetration, but at the expense of reduced motion flexibility, occasionally hindering precise goal adherence.

**Does consistency model help updating scene state?** The comparison between the second and the last rows of Table 3 shows that replacing the distilled consistency model (CM) with a standard diffusion model causes a drastic drop in generation speed, making iterative refinement computationally infeasible for real-time applications. In contrast, the CM's few-step generation capability makes efficient iteration possible. Further, the quality of the previous sample used for scene state updates is paramount. The CM yields a higher-quality motion estimate, which provides a more accurate basis for updating the dynamic scene state. This leads to better scene awareness, evidenced by the reduction in both human penetration and object penetration with scene. These findings confirm that the consistency model is not just an accelerator; its ability to provide high-quality, few-step motion predictions is essential for reliable and effective dynamic scene state updating.

## 5 Conclusion

This paper presents INFBAGEL, a novel framework for human-object-scene interaction generation that addresses the critical challenges of dynamic object–scene changes and data scarcity. Our coarse-to-fine approach utilizes the few-step denoising of a consistency model, achieving dynamic perception of the scene. To enhance physical plausibility, we introduce a lightweight bump-aware guidance that effectively guides the model toward collision-free optimization. Furthermore, our hybrid data training strategy combines real-scene HSI data with synthesized HOI data, effectively overcoming data limitations and enabling zero-shot seen generalization.

ACKNOWLEDGMENTS

This work was supported in part by the National Natural Science Foundation of China under Grant 62401367 and in part by Shanghai Artificial Intelligence Laboratory.

## REPRODUCIBILITY STATEMENT

We are committed to ensuring the reproducibility of this work. To this end, we will publicly release the complete source code, pre-trained models, and the HOSI (Human-Object-Scene Interaction) benchmark dataset that we have constructed after the paper is published. Details about the used dataset, including the hybrid data processing pipeline for converting HOI data into HOSI tuples, as well as the construction details of the HOSI benchmark, are described in detail in Appendix A.5. Our model architecture, including the specific implementation of the dynamic perception encoder and the motion consistency model, is elaborated in Section 3 of the methodology, and more detailed implementation parameters and network structures are provided in Appendix A.6. Finally, all training hyperparameters, inference settings, including iterative optimization and collision-aware guidance parameters can be found in Appendix A.7.

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

## A  APPENDIX

### A.1  MORE QUALITATIVE RESULTS

More qualitative results are presented in Fig. 4, with more scenes, object types and motion types, including lifting over head, kicking and interaction with static scene.

As shown in Fig. 5, INFBAGEL is largely environment-independent because it conditions on a voxelized scene representation, rather than relying on detailed scene geometry with a fixed scene category. As long as a new environment, whether a household, store, or hospital, can be voxelized, the same network can be applied.

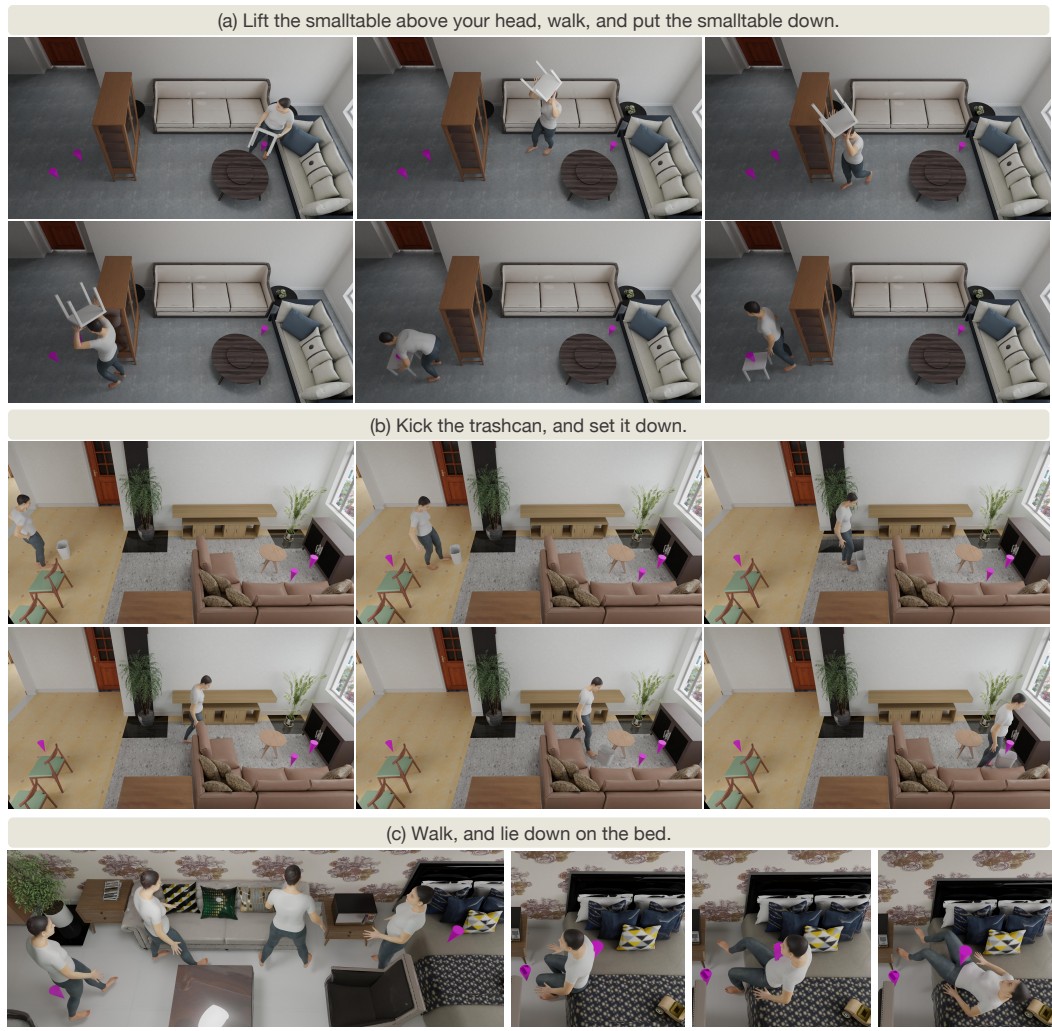

Figure 4: Qualitative results on different scenes, motion types and object types. The top two rows (a/b) show diverse human-object interactions including lifting over head and kicking. The last row shows a static scene interaction.

## A.2 HOI EXPERIMENT

**Experiment Settings.** To assess INFBAGEL against specialized HOI methods, we evaluate on the standard HOI benchmark OMOMO. Following CHOIS, we assess the results from multiple perspectives. For condition matching, we adapt the metrics to our condition, measuring the object's final position error ($T_e$) and the human waypoints error ($T_{xy}$). Human motion quality is evaluated using foot sliding (FS), Fréchet Inception Distance (FID), and R-precision ($R_{prec}$) to measure physical plausibility and text-motion consistency. For interaction quality, we use contact precision, recall, F1-score ($C_{prec/rec/f1}$) and contact percentage ($C_\%$), and measure the penetration between the human body and the object ($P_{body}$). Finally, the difference from ground truth is quantified by the mean per joint position error (MPJPE), the root translation error ($T_{root}$) and the object pose error ($T_{obj}, O_{obj}$). There are a large number of noise points in the SDF of some objects in OMOMO, which makes the calculation of $P_{body}$ on the entire sequence meaningless. Therefore, we skipped the calculation of $P_{body}$ on noise objects, including *woodchair*, *whitechair*, *largebox*, *largetable*, *plasticbox* and *trashcan*. For the unified penetration measure, we calculate $P_{body}$ by summing the total depth of penetration from any frame through vertices, rather than the average value, measured in meters (m).

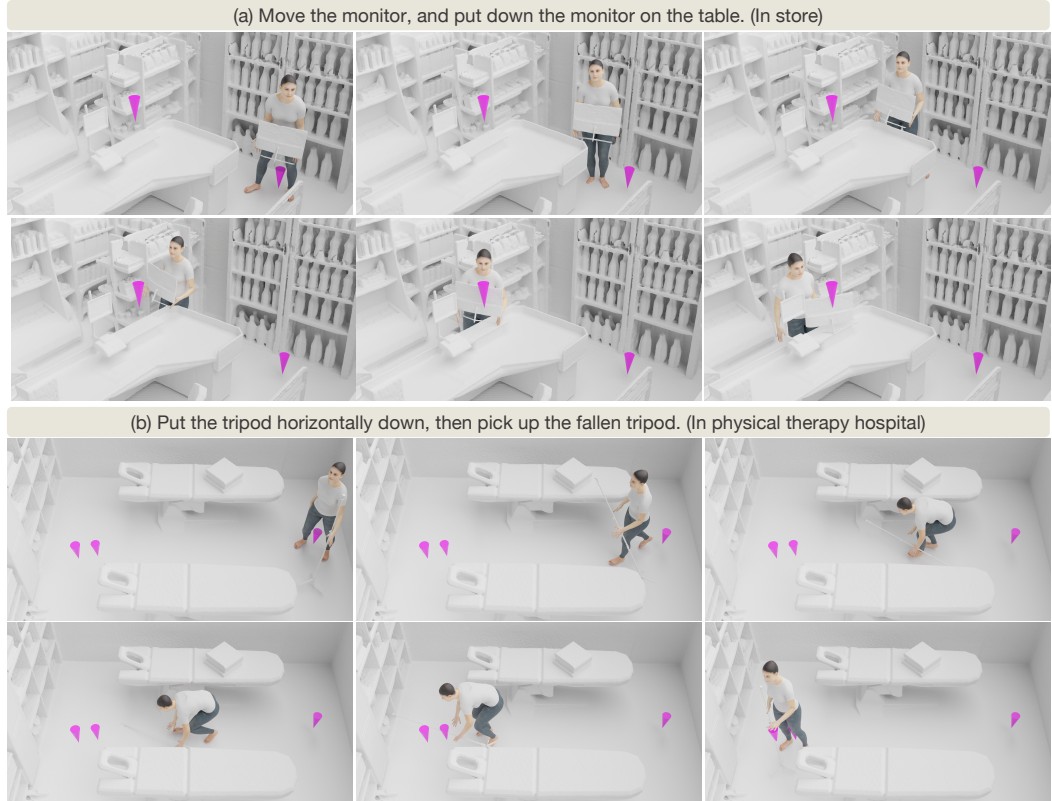

Figure 5: Qualitative results in socially-interactive scenes, including a store and a physical therapy room. These unseen scenes are chosen from LINGO, displayed in default white due to the lack of texture.

**Baseline Comparison.** We compare INFBAGEL and LINGO (Jiang et al., 2024a), by treating them in an empty scene, including CHOIS (Li et al., 2024b) and ROG (Xue et al., 2025), two standard methods for HOI generation.

CHOIS generates synchronized human and object motion from a language description, initial states, and sparse object waypoints. To ensure realism, it utilizes an object geometry loss to better match the generated motion to the waypoints and incorporates guidance terms to enforce hand-object contact constraints during sampling. To align it with our method, we modify it to be conditioned on human waypoints and a object goal.

ROG generates human-object interactions using rich geometric detail. It selects key points from the object's mesh to create a detailed geometric representation and constructs an interactive distance field (IDF) to capture interaction dynamics. A diffusion-based relation model with spatial and temporal attention mechanisms is used to guide the motion generation process, ensuring movements are relation-aware and semantically correct. Similarly, we adapt ROG conditioning to add human waypoints and a object goal.

We evaluate INFBAGEL with different iteration steps (1, 8, and 16) with the guidance of CHOIS, to demonstrate the effectiveness of our iterative optimization. Competitive results can be achieved through simpler guidance and fewer optimization steps (CHOIS and ROG used 10 optimization steps).

**Results on HOI.** Table 5 presents the quantitative results on the OMOMO dataset. INF-BAGEL consistently outperforms all baselines across most metrics. A key advantage of our method is its exceptional efficiency. With just a single sampling step (INFBAGEL $_1$), our model achieves performance comparable to CHOIS across several key metrics (e.g., $C_{f1}$ of

0.66), while operating at an astonishing speed of over 1500 FPS. This highlights the high quality of the initial generation from our consistency model.

Furthermore, the results clearly demonstrate the benefit of our iterative optimization. As the number of sampling steps increases from 1 to 16, we observe a significant and consistent improvement in motion and interaction quality. Specifically, INFBAGEL $_{16}$ surpasses all baselines, achieving the best scores in FID, contact quality ($C_{rec}$, $C_{f1}$, $C_{\%}$), and human-object penetration ($P_{body}$), all while maintaining a real-time generation speed (29.31 FPS) comparable to CHOIS. This confirms that our framework not only provides a high-quality initial estimate but also effectively refines it toward more physically plausible and accurate interactions, showcasing a superior balance of speed and quality.

Table 5: Performance Evaluation on the HOI Dataset.

| Method | Condition Matching↓ | | Human Motion | | | Interaction | | | | | GT Difference↓ | | | | FPS↑ |
|---|---|---|---|---|---|---|---|---|---|---|---|---|---|---|---|
| | $T_e$ | $Txy$ | $FS\downarrow$ | $R_{prec}\uparrow$ | $FID\downarrow$ | $C_{prec}\uparrow$ | $C_{rec}\uparrow$ | $C_{f1}\uparrow$ | $C_{\%}\uparrow$ | $P_{body}\downarrow$ | $MPJPE$ | $T_{root}$ | $T_{obj}$ | $O_{obj}$ | |
| Real Motion | 0.00 | 0.00 | 0.26 | 0.73 | 0.00 | - | - | - | 0.66 | 1.29 | 0.00 | 0.00 | 0.00 | 0.00 | - |
| LINGO | 17.57 | 3.76 | 0.50 | 0.58 | 2.57 | 0.70 | 0.46 | 0.50 | 0.41 | 4.80 | 16.39 | 10.28 | 27.64 | 1.33 | 28.86 |
| CHOIS | 5.28 | 5.73 | **0.32** | **0.68** | 1.01 | 0.78 | 0.63 | 0.66 | 0.53 | 2.52 | 13.93 | 8.42 | 17.74 | **0.92** | 28.01 |
| ROG | 26.68 | 19.43 | 0.49 | 0.53 | 5.46 | **0.80** | 0.68 | 0.70 | 0.54 | 5.86 | 13.37 | 21.28 | 31.89 | 1.14 | 9.07 |
| INFBAGEL $_1$ | **2.93** | 4.19 | 0.36 | 0.63 | 3.41 | 0.79 | 0.64 | 0.66 | 0.52 | 2.83 | **11.92** | 8.88 | **15.40** | 1.02 | **1566.71** |
| INFBAGEL $_8$ | 2.99 | 3.89 | 0.34 | 0.67 | 1.78 | 0.78 | 0.68 | 0.70 | 0.56 | 2.61 | 11.97 | 8.47 | 15.59 | 1.02 | 60.94 |
| INFBAGEL $_{16}$ | 3.06 | **3.70** | **0.32** | 0.67 | **0.68** | 0.78 | **0.73** | **0.73** | **0.60** | **2.49** | 12.11 | **7.93** | 15.93 | 1.02 | 29.31 |

## A.3 ABLATION STUDY ON HYBRID DATA RATIO

To investigate the impact of different ratios of synthesized OMOMO data to LINGO data in our hybrid data training strategy, we conducted an ablation study by training INFBAGEL with three different ratios: 1:0 (only synthesized OMOMO data), 1:0.5, and 1:1. The evaluation results are presented in Table 6.

The results show that the gain from HSI data on scene understanding has a limit, too much HSI data may compromise the model's ability to learn object manipulation from HOI data, indicating a trade-off between scene-level physical plausibility and task-specific interaction priors. A balanced ratio of 1:1 or 1:0.5 between synthesized OMOMO data and LINGO data achieves the best overall performance across all metrics.

Table 6: Ablation study on the different ratios of synthesized OMOMO data to LINGO data.

| Method | Task Accuracy | | | FS↓ | HO Interaction | | HS Penetration↓ | | | OS Penetration↓ | | |
|---|---|---|---|---|---|---|---|---|---|---|---|---|
| | $T_h\downarrow$ | $T_o\downarrow$ | $S_{\%}\uparrow$ | | $C_{\%}\uparrow$ | $P_{body}\downarrow$ | $P_{mean}$ | $P_{max}$ | $P_{f\%}$ | $P_{mean}$ | $P_{max}$ | $P_{f\%}$ |
| INFBAGEL (1:0) | 4.75 | 8.14 | **83.16** | **0.13** | **78.18** | **3.96** | 3.39 | **36.97** | 28.19 | 16.62 | 109.61 | 22.72 |
| INFBAGEL (1:0.5) | **4.37** | **7.94** | 81.45 | 0.15 | 76.96 | 5.05 | 3.17 | 37.09 | **26.74** | **12.45** | **93.37** | **22.32** |
| INFBAGEL (1:1) | 4.8 | 9.44 | 69.72 | 0.18 | 76.48 | 4.01 | **3.13** | 39.39 | 27.87 | 16.00 | 105.76 | 23.94 |

## A.4 EXTENDED RELATED WORK

### A.4.1 INTERACTION WITH EVERYTHING

**Interaction with Small Objects.** Early research primarily focused on synthesizing hand motions (Zhang et al., 2021; Christen et al., 2022; Paschalidis et al., 2025; Zhang et al., 2025b). With the emergence of motion datasets encompassing full-body hand-object interactions (Fan et al., 2023; Taheri et al., 2020; Lv et al., 2024), the focus expanded to synthesizing full-body motions for object grasping (Taheri et al., 2022; Wu et al., 2022). Some methods (Braun et al., 2024; Ghosh et al., 2023; Li et al., 2024c) simultaneously generate human motion and predict object motion; however, these approaches are mainly limited to interactions with small objects, with an emphasis on hand motion.

**Interaction with Large Objects.** For interaction with large objects, the complexity of human motion increases, and object motion becomes non-negligible. Some methods employ reinforcement learning strategies to synthesize actions for specific tasks, such as box moving (Merel et al., 2020) or basketball sport (Liu & Hodgins, 2018; Wang et al., 2025a). Recent mimic learning methods (Xu et al., 2025; Yu et al., 2025) have achieved generalized character interaction. With the growing availability of human-object interaction datasets (Bhatnagar et al., 2022; Li et al., 2023; Lu et al., 2025b), certain methods (Li et al., 2024b; Cong et al., 2025) have started generating motions for interactions with large objects, often relying on sequential points or object trajectories, which restricts the model's ability to autonomously generate diverse interactions. Other methods (Xu et al., 2023; Diller & Dai, 2024; Song et al., 2024; Peng et al., 2025; Li et al., 2025; Xue et al., 2025; Zeng et al., 2025) attempt to simultaneously synthesize human and object motions but require additional models for optimization, unable to achieve real-time generation.

**Limitation of HOI dataset.** Due to the general lack of scene annotations in datasets, human-object interactions are typically placed into scenes by plan collision-free paths (Li et al., 2024b; Wu et al., 2024). However, these methods lacks direct perception awareness, limiting their applicability in complex and dynamic 3D environments.

### A.4.2 INTERACTION IN EVERYWHERE

**Static Interaction in Scene.** Early work primarily focused on generating interactions within static scenes, extending from locomotion in the scene (Zhang & Tang, 2022; Mao et al., 2022) to interacting with static objects (Zhang et al., 2020b; Xuan et al., 2023; Zhao et al., 2022; Wang et al., 2022b), such as sitting and lying down. Subsequently, some methods considered both locomotion and static interaction simultaneously. Hassan et al. (2021), Wang et al. (2022a), Huang et al. (2023), Mir et al. (2024) and Zhang et al. (2024) modeled them separately, resulting in inconsistent motions. Huang et al. (2023), Zhao et al. (2023), Yi et al. (2024) and Lou et al. (2024) relied on high-level path planners for locomotion, making sensitivity to the quality of the planner. Zhang et al. (2020a), Wang et al. (2024b) and Cen et al. (2024) are multi-stage frameworks, which incurred significant computational overhead. Recently, Some methods (Zhao et al., 2025; Wang et al., 2025b; Chen et al.) based on LLMs have shown strong generalization ability in static HSI.

**Dynamic Interaction in Scene.** Recently, interactions with dynamic objects in the scene gains increasing attention. Some physics-based methods (Lee & Joo, 2023; Hassan et al., 2023; Pan et al., 2024; Xu et al., 2024; Xiao et al., 2024; Deng et al., 2025; Pan et al., 2025; Wang et al., 2024a; Gao et al., 2024; Zhang et al., 2025a) explore the interactions between physical simulated humanoids and dynamic objects through reinforcement. However, these methods often require complex reward functions engineering and still struggle with motion diversity and realism. Kinematics-based methods Jiang et al. (2024b) and Jiang et al. (2024a) propose a unified generation framework, but simplify object motion, by simply attaching to the hand. Li & Dai (2024) and Li et al. (2024a) achieve generalized generation by the powerful prior of image or video generation models, but is limited by time-consuming generation and complex post-processing. Our most similar work are Yao et al. (2025) and Geng et al. (2025), which generate object motion. However, these methods only employ a one-time static scene awareness and lack object perception of the scene, also restricted by the limited object types, text, and scene annotations in the dataset.

**Limitation of HSI dataset.** Due to the high cost of data collection, current scene-annotated datasets are limited. Liu et al. (2024) aggregate motion-only data into paired human-occupancy interaction data, substantially improving scene diversity. Our method is not simply an extension, but further combines enhanced HOI and HSI data on the basis of this method, decoupling the data required for the HOSI task through a hybrid training strategy. Recently, some work introduces object interactions in scene. Jiang et al. (2024b) includes object motion but lacks instruction-level annotations, and the variety of objects is restricted. Jiang et al. (2024a) features diverse scenes and text annotations but omits object motion. Kim et al. (2025) has a limited variety of scenes, making it difficult to achieve generalized scene perception. The restricted variety of objects and scenes limits perception and generation for dynamic interactions in complex scenes.

### A.4.3 Generation All at Once

Diffusion-based acceleration methods (Song et al., 2020; Lu et al., 2022; 2025a) have demonstrated strong capabilities in various domains, such as video generation (Wang et al., 2023), 3D reconstruction (Ye et al., 2024), and Embodied AI (Lu et al., 2024; Zhu et al., 2025). Consistency models achieve high-quality samples with few-step sampling and also can be optimized through iterative generation (Song et al., 2023; Luo et al., 2023), introduced into the field of motion generation by Dai et al. (2024) recently, but the aspect of interaction generation still awaits exploration. We use the inherent nature of consistency models to achieve a unified framework with dynamic perception through few-step generation, and a bump-aware sampling guidance for iterative optimization, while improving efficiency.

### A.5 Datasets and Benchmark Details

**Training Datasets.** Our hybrid data training strategy utilizes two existing high-quality datasets: (1) LINGO (Jiang et al., 2024a): This is a large-scale human-scene interaction (HSI) dataset that provides diverse indoor scenes (high-fidelity meshes), rich text instructions, and corresponding 3D human actions. We use it to train the model's navigation and interaction capabilities in complex static environments. (2) OMOMO (Li et al., 2023): This is a large-scale human-object interaction (HOI) dataset that includes full-body motion capture data for interacting with various manipulable objects (especially large objects). We use it to learn fine, physically plausible human-object interaction dynamics.

Due to the lack of scene information in the OMOMO dataset, we adopted a synthetic strategy to convert it into a human-object-scene (HOSI) tuple suitable for our unified framework. The process follows the method of Liu et al. (2024), with the specific steps as follows:

For each motion sequence in OMOMO, we first calculate the union of all space points occupied by the human (via the SMPL-X mesh) and the manipulated object throughout the entire time series. We create a bounding box around this dynamic occupancy volume and voxelize the space outside the bounding box, marking it as non-traversable "walls" or "obstacles." This synthesizes a minimal but logically reasonable scene context for the HOI data without a scene, enabling the model to learn basic spatial obstacle avoidance constraints during training. The synthesized data is processed into the same $(\mathcal{S}, \mathcal{O}, \mathcal{T}, \mathcal{G})$ format with the LINGO dataset, where $\mathcal{S}$) is the synthesized voxelized scene, $\mathcal{O}$) is the object geometry, $\mathcal{T}$) is set according to the original data, and $\mathcal{G}$) is set to the pose of the motion endpoint.

**HOSI Benchmark Construction.** In order to comprehensively evaluate the model's generalization ability on complex, unseen HOSI tasks, we have constructed a new benchmark. We selected 67 diverse indoor scenes from the TRUMANS dataset and excluded the objects with dirty SDFs from the OMOMO dataset, choosing 7 categories of large manipulable objects (such as boxes, chairs, tables, etc.). In each scene, we randomly select start and goal position for each object type, and choose a pair of initial and target poses from the OMOMO-test including human and object, ensuring that both initial and goal poses have no collision in the scene. We ensure that there is a passage between the start and target points, but the path may need to detour. The initial motion and text instruction are sampled from OMOMO-test and therefore inherits diverse motion types, including lifting, kicking, dragging, rotating and so on, ensuring comprehensive and representative evaluation. By combining different scenes, objects, starting points, and target points, we have collectively generated 469 independent test sequences, each of which constitutes a unique HOSI task.

### A.6 Model Architecture

**Motion Generator.** Our motion generator is based on a Transformer encoder architecture, which includes 8 layers and 16 attention heads. Each frame's raw motion state of the human and interactive object is first embedded into a 512-dimensional latent space via an MLP, where the diffusion process is applied. The resulting noised motion tokens, along with conditional tokens (scene, text, goal and object geometry), are fed into the Transformer for denoising. The denoised latent tokens are finally decoded by another MLP to reconstruct the motion state in the original space. Other conditions are embedded using the corresponding

encoder. An embedding of diffusion timestep is added to the condition tokens and all tokens undergo a positional encoding before being fed into the Transformer. It generates motion sequences in an autoregressive manner, by replacing the first two tokens of the current window to the last two frames from the previous window without noise.

**Scene Perception Encoder.** We use a Vision Transformer (ViT) architecture (Dosovitskiy et al., 2020) with 6 Transformer layers and 16 attention heads to encode scene features from 3D voxel inputs. The input consists of five N×N×N voxel grids (3 dynamic and 2 static), which are flattened and processed into a patch sequence, with N=32 denoting the number of voxels per spatial dimension. Each 32×32×32 grid represents a cubic region of 1.2 meters per side centered at the pelvis/goal, so each element corresponds to a 3.75 centimeter cube indicating occupancy or emptiness. The ViT outputs a 512-dimensional embedding for each voxel grid, which is further fused with the positional encoding of its corresponding pelvis or goal position to incorporate precise spatial cues. These five 512-dimensional embeddings jointly constitute the scene condition.

**Text Encoder.** First, we use the CLIP encoder (Radford et al., 2021) to convert the textual instruction into encoding with 768-dimension. Then, we use an MLP to transform this encoding into 512-dimension as the final text condition.

**Goal Encoder.** We use different MLPs to encode pelvis and object goals into two 512-dimensional vectors. For the pelvis goal, we retain only the 2-dimensional horizontal coordinates as input for the model. For the object goal, the 3-dimensional target coordinates serve as input for the model. For motion that do not involve object, we mask the object goal tokens as zeros.

**Object Geometry Encoder.** We use an MLP to project the object BPS representation from the 1024×3 vector to a 512-dimensional embedding, which serves as the object geometry condition.

## A.7  IMPLEMENTATION DETAILS

**Training Details.** We used the Adam optimizer for training, with a learning rate set to 0.0001, and a batch size of 2048. The model was trained for about 500 epoches on 4 NVIDIA A100 GPUs. During training, We randomly mask the dynamic scene encoding with a ratio of ten percent. For distillation, we use DDIM as the ODE solver with total 25 timesteps. The scale $\omega$ of classifier-free guidance (CFG) is uniformly sampled from the interval $[0, 3.0]$ during training. The weight settings for the total loss function are $\lambda_h = 1$ and $\lambda_o = 0.5$ to balance the consistency loss and the kinematic auxiliary loss.

**Inference Details.** In reasoning, we first use 1 step to generate a rough motion trajectory (without using dynamic scenes as a condition). Then, we perform 15 steps for optimization with guidance. In each iteration, we use the actions generated in the previous iteration to update the dynamic scene voxels. Bump-aware guidance is activated in each sampling iteration. The guidance scale $\gamma_{\tau_n}$ is related to the sampling step $\tau_n$, providing stronger guidance in the early stages of sampling. Pre-computation of the distance map makes the additional overhead of this module extremely minimal.

## A.8  LARGE LANGUAGE MODEL USAGE STATEMENT

We used the large language model as a general-purpose language-polishing tool. The model was prompted to improve grammar, word choice, and sentence fluency of selected portions of the manuscript. All generated suggestions were manually reviewed and edited by the authors, who take full responsibility for the final text.

