# OpenReview forum: "InfBaGel: Human-Object-Scene Interaction Generation with Dynamic Perception and Iterative Refinement"
_ICLR.cc/2026/Conference — ICLR 2026 Poster_

### Official Review · Reviewer_73R8 · 2025-10-27

**Soundness:** 3
**Presentation:** 2
**Contribution:** 3
**Rating:** 6
**Confidence:** 2

**Summary:**

This paper presents InfBaGel, a unified framework for human–object–scene interaction generation that combines a coarse-to-fine consistency model with bump-aware guidance for physically plausible motion (mostly for collision free). It also introduces a hybrid data training strategy that mixes real and synthetic data to reduce annotation needs and improve generalization across diverse scenes.

**Strengths:**

1. The motivation and contribution is clear: to generate higher quality HOSI data and to tackle the data scarcity of this field.
2. Experiment shows that the method is significantly better than previous works in terms of collision avoidance.

**Weaknesses:**

1. Limited qualitative results. Only two action sequences are presented in the paper, which makes it difficult to fully assess the model’s performance.
2. The method part confuse me a bit. Specifically, for Section 3.2 (“Motion Consistency Model”), does this section correspond to the Auto-regressive Consistency Model shown in the figure? Additionally, in line 277, how are human joints and object points constructed from human motion only? Is this perhaps a typo?
3. In Figure 2(c), there is a noticeable penetration between the hand and the box. Is this due to training solely on the OMOMO dataset?

**Questions:**

See weaknesses.

---

> ### Author Response · Authors · 2025-11-21
> **Official Comment by Authors**
>
> Thank you for your thoughtful comments and questions.
>
> > Limited qualitative results. Only two action sequences are presented in the paper, which makes it difficult to fully assess the model’s performance.
>
> In the revision, we have added a comparison of dragging motion example in Figure 2 , and a pushing motion example in Figure 3. Due to space limitations, we further provide motion like lifting over the head, kicking and so on in Figure 4 and Figure 5 of Appendix A.1 across different scenes and object types. These examples demonstrate the robustness and applicability of the proposed method across diverse instructions and scenes.
>
> > The method part confuse me a bit. Specifically, for Section 3.2 (“Motion Consistency Model”), does this section correspond to the Auto-regressive Consistency Model shown in the figure?
>
> Thank you for pointing out the confusion. To clarify, the "Motion Consistency Model" in Section 3.3 does not correspond to the "Auto-regressive Consistency Model" mentioned in the figure. To address this, we have revised the Section 3.3 (“Motion Consistency Model”)  to improve clarity and better explain the methodology.  After revision, this section is organized as follows. (1) We first describe how the diffusion model performs coarse-to-fine generation with dynamic perception.  (2) We then explain the motivation for distilling the multi-step diffusion model into a few-step consistency model, which improves both inference speed and dynamic scene perception. (3) Finally, we present the objective used to achieve consistency distillation. Furthermore, Fig 1(a), illustrating the architecture of the motion model, is renamed as Auto-regressive Motion Model to avoid confusion.
>
> > Additionally, in line 277, how are human joints and object points constructed from human motion only? Is this perhaps a typo?
>
> Thank you for the correction. It is a typo. The correct text should be: "the consistency model predicts the clean data $\hat{\mathbf{x}}_{0}$ which derive $\hat{\mathcal{M}_h}$ and $\hat{\mathcal{M}_o}$ to reconstruct the human joints and object points." We have corrected it in the revision.
>
> > In Figure 2(c), there is a noticeable penetration between the hand and the box. Is this due to training solely on the OMOMO dataset?
>
> Yes, we agree that due to the lack of hand annotations in OMOMO [1] dataset and the artifacts introduced during the motion capture process, achieving perfect contact and zero penetration base on OMOMO in a purely kinematic generation setting is challenging.
>
> Notably, compared with the kinematics‑based baselines [2, 3, 4, 5], our approach substantially improves physical feasibility, achieving the lowest human-object penetration and highest contact rates on both HOSI and HOI benchmarks (as shown in the following Table,  from Table 1 and Table 5 in the revised paper), demonstrating superior physical realism.
>
> | Method                                 |      C% ↑ | P_body ↓ |
> | -------------------------------------- | --------: | -------: |
> | *Comparision on the HOSI benchmark* |           |          |
> | TRUMANS [2]                                |     0.46 |     4.91 |
> | LINGO [3]                                 |     0.54|     5.72 |
> | Ours                               | **0.78** | **3.96** |
> | *Comparision on the OMOMO benchmark*|           |          |
> | LINGO [3]                                  |     0.41 |     4.80 |
> | CHOIS [4]                                 |     0.53 |     2.52 |
> | ROG [5]                                    |     0.54 |     5.86 |
> | Ours                                 | **0.60** | **2.49** |
>
> *[1] Object Motion Guided Human Motion Synthesis. Li et al.*
>
> *[2] Scaling Up Dynamic Human-Scene Interaction Modeling. Jiang et al.*
>
> *[3] Autonomous Character-Scene Interaction Synthesis from Text Instruction. Jiang et al.*
>
> *[4] Controllable Human-Object Interaction Synthesis. Li et al.*
>
> *[5] Guiding Human-Object Interactions with Rich Geometry and Relations. Xue et al.*

---

### Official Review · Reviewer_f1br · 2025-10-31

**Soundness:** 3
**Presentation:** 3
**Contribution:** 3
**Rating:** 6
**Confidence:** 4

**Summary:**

The problem this paper aims to tackle is the (1) limited annotation in dataset and the (2) dynamic scene variation (the movement of objects changes the scene as well), for a task termed as human-object-scene interaction generation. The proposed approach consists of several contributions/effective modules, termed as the dynamic perception, bump-aware guidance, the consistency guided iterative refinement, with the help of hybrid-data-training strategy. It seems that using the proposed method achieves good results in the new benchmark HOSI and the HOI as well.

**Strengths:**

Strength
-	The technical method seems correct
-	The motivation is well-presented

**Weaknesses:**

Weakness

-	In table 3, it seems that the C+B provides inferior results. It does not convince me that all components achieve better performance, esp in the task accuracy metric. Maybe more explanations should be provided.

-	It seems that the baseline methods in comparison seem limited. More evaluations should be performed comparing with more methods.


-	The proposed contributions seem a lot. These makes the contributions diluted for some perspectives. The relationship among these contributions, and evaluation of their effectiveness one by one is necessary. Otherwise we cannot understand how each component tackles which portion of data.

**Questions:**

See weakness

---

> ### Author Response · Authors · 2025-11-21
> **Official Comment by Authors (1/2)**
>
> Thank you for your thoughtful comments and questions.
>
> > In table 3, it seems that the C+B provides inferior results. It does not convince me that all components achieve better performance, esp in the task accuracy metric. Maybe more explanations should be provided.
>
> The results in Table 3 reflect a trade-off between task accuracy (goal-reaching) and physical plausibility (penetrations). Bump-aware guidance (B) encourages the model to avoid collisions with the scene, which guides the generator to choose a safer, collision-free path although slightly deviates from the given goal position.
>
> As a result, the final position may fall outside the success threshold used in our task accuracy metric, even though the trajectory is physically more reasonable. This behavior is consistent with our description in ablation study, where we note that stricter collision constraints enhance physical plausibility by mitigating penetration, but at the expense of reduced motion flexibility, occasionally hindering precise goal adherence.
>
> In other words, C+B does not harm the model in a random way; it shifts the solution toward collision-aware, safe motions, which improves penetration scores at the cost of a small drop in task accuracy. In many downstream applications (e.g., robotics or animation in cluttered scenes), collision alleviation is often more critical than marginal gains in goal accuracy. We have added a discussion about the trade-off in Sec. 4.3 (L. 514-518) in the revision.
>
> > It seems that the baseline methods in comparison seem limited. More evaluations should be performed comparing with more methods.
>
> We agree that broader comparisons are important. However, HOSI is a new problem setting, and closely related methods is currently quite limited, due to the lack of large-scale HOSI datasets. To the best of our knowledge, only two concurrent works, HOSIG [1] and UniHM [2], explore similar domains.  We have discussed the technical differences and limitations in the Related Work Section.
>
> Concretely, HOSIG considers limited motion and object types in HOSI. UniHM matches OMOMO motions to collision-free scenes, but unmatched cases are treated as empty scenes. Both of them still adopt static scene perception and depend on external path planners. In contrast, InfBaGel considers diverse motion types and scenes with the proposed hybrid data training strategy. It is designed as a unified coarse-to-fine framework with few-step  consistency sampling and dynamic perception,  enabling more adaptive and realistic HOSI interactions.
>
> Since both are preprints without released code, a fair quantitative comparison is currently infeasible. Given these constraints, we compare against the state-of-the-art baselines they adopt, covering the two key dimensions our task covers. Specifically, on the HSI/HOSI side, we compare to voxel-based SOTA human-scene interaction models TRUMANS and LINGO on our HOSI benchmark, showing clear gains in the success rate and physical feasibility. On the HOI side, we compare to SOTA human-object interaction models CHOIS and ROG, demonstrating that our approach remains competitive in motion quaility and interaction accuracy. Together, these results demonstrate clear advantages of our framework in dynamic scenes understanding and interaction generation, establishing a strong baseline for future HOSI research.
>
> *[1] HOSIG: Full-Body Human-Object-Scene Interaction Generation with Hierarchical Scene Perception. Yao et al.*
>
> *[2] UniHM: Universal Human Motion Generation with Object Interactions in Indoor Scenes. Geng et al.*
>
> > The proposed contributions seem a lot. These makes the contributions diluted for some perspectives. The relationship among these contributions, and evaluation of their effectiveness one by one is necessary. Otherwise we cannot understand how each component tackles which portion of data.
>
> Thanks for the advice. We have polished the Introduction Section to make the contrition more clear, as highlighted in blue in the revision.  It is characterized by
> - A coarse‑to‑fine conditional interaction generation framework aligned with the few‑step denoising of a consistency model, in which a dynamic perception strategy adaptively updates scene context and a bump‑aware guidance further improves physical plausibility.
> -  A hybrid data training strategy that combines synthetic HOSI data with high-fidelity HSI data, relaxing the reliance on complete HOSI annotations and addressing data scarcity.
>
> In addition, we have reorganized the ablation study in the Experiment Section to evaluate their effectiveness one by one, where each contribution is evaluated via a targeted ablation that either removes or replaces that component to isolate its effect.  Furthermore, we have added a visual comparison of different ablationed variants in Figure 3 in the revision. Below we clarify the ablation design and summarize the per-component analysis.

---

> ### Author Response · Authors · 2025-11-21
> **Official Comment by Authors (2/2)**
>
> Specifically, (1) to examine the effect of **dynamic perception encoding (DP)**, we compare a model using only static scene encoding (i.e., without DP) with one adopting DP, as presented in Table 3, and further explore the impact of the number of dynamic voxels, as shown in Table 4. (2) We further study the effect of **bump-aware guidance (B) and contact guidance (C)** on improving physical plausibility, represented as DP+C and DP+C+B in Table 3. (3) Next, we compare the distilled **consistency model** with a standard diffusion model, both using DP, as shown in Table 3, and study the influence of sampling steps in Table 4. (4) Finally, we compare the models trained on expanded-OMOMO solely with the models trained on the **hybrid dataset**. Qualitative comparisons of different variants are provided in Figure 3.
>
> 1. For Dynamic Perception Encoder (DP):  Comparing the first two rows of Table 3 reveals the impact of our dynamic perception encoding. Disabling this component (row 1) leads to a notable degradation across multiple metrics compared to the model with DP enabled (row 2). In particular, scene penetration by both the human and the object increases, indicating that without dynamically updating its perception of the scene, the model becomes less capable of avoiding collisions and  performing fine‑grained interaction. As shown in Figure 3 (b) and (d), only the static scene encoding results in penetration between the human and scene, while this does not occur when dynamic scene encodings are included. In addition, the overall task success rate drops sharply from 86.35% to 71.22%, and foot sliding worsens from 0.14 to 0.19. This observation is further supported by the results in Table 4. Reducing the temporal voxel number from three to one causes a decline in success rate and worse motion quality in terms of foot sliding. These results underscore that dynamic perception is crucial for generating physically plausible, goal-directed motions in complex and evolving environments.
>
> 2. For Bump-aware Guidance (B): As shown in Table 3 (rows 2–4), introducing contact guidance (C) leads to notable improvements in human-object (HO) interaction metrics, including $C_{\\%}$ and $P_{body}$. Furthermore, incorporating the proposed bump‑aware guidance (B) yields consistent and substantial improvement in both HS and OS penetration metrics, while maintaining the HO interaction performance nearly unchanged. Figure 3 (c) and (d) further demonstrate the necessity of bump‑aware guidance. These results validate the effectiveness of bump‑aware guidance in enhancing the physical plausibility of human-scene and object-scene interactions. However, we also observe that there is a slight degradation in task accuracy metrics such as the goal distance ($T_h$ and $T_o$), which highlights a trade‑off introduced by the guidance. Stricter collision constraints enhance physical plausibility by mitigating penetration, but at the expense of reduced motion flexibility, occasionally hindering precise goal adherence.
>
> 3. For Consistency Model vs. Diffusion Model (CM vs. DM): The comparison between the second and the last rows of Table 3 shows that replacing the distilled consistency model (CM) with a standard diffusion model causes a drastic drop in generation speed, making iterative refinement computationally infeasible for real-time applications. In contrast, the CM's few-step generation capability makes efficient iteration possible. Further, the quality of the previous sample used for scene state updates is paramount. The CM yields a higher-quality motion estimate, which provides a more accurate basis for updating the dynamic scene state. This leads to better scene awareness, evidenced by the reduction in both human penetration and object penetration with scene. These findings confirm that the consistency model is not just an accelerator; its ability to provide high-quality, few-step motion predictions is essential for reliable and effective dynamic scene state updating.
>
> 4. For Hybrid Data Training Strategy: As shown in the second group of Table 1, both models trained on hybrid data achieve significantly reduced human and object penetration with scene. This indicates that incorporating HSI data, containing high-fidelity scene geometry, enhances the model's understanding of complex environmental constraints and improves its ability to avoid collisions while navigating the object in scenes. These results demonstrate that our hybrid data strategy contributes to learning generalizable scene representation, enabling the model to perform complex human-object-scene Interaction tasks effectively in unseen environments. Notably, the task type in the LINGO data is different from the HOSI test set, making a slight trade-off in success rate and human-object interaction metrics ($C_{\\%}$ and $P_{body}$), this is expected as the model learns to prioritize scene-level physical plausibility over strictly adhering to interaction priors learned from HOI data.

---

### Official Review · Reviewer_9B2s · 2025-10-31

**Soundness:** 4
**Presentation:** 3
**Contribution:** 3
**Rating:** 6
**Confidence:** 4

**Summary:**

The paper introduces a coarse-to-fine, instruction-conditioned generation framework using a consistency model that iteratively refines the human-object-scene interaction (HOSI).

A dynamic perception strategy is key, where scene context is continuously updated using trajectories from the preceding refinement step to condition the subsequent refinement, ensuring consistent interactions.

Experiment results show that the hybrid data training strategy overcomes data limitations by combining
real-scene HSI data with synthesized HOI data, achieving zero-shot scene generalization.

**Strengths:**

The core idea is to integrate a dynamic perception strategy and a coarse-to-fine iterative refinement scheme, which is aligned with a consistency model's denoising process.

This novel approach ensures that the scene context (which changes due to object and human movement) is updated at each step, leading to more physically consistent and realistic HOSI.

Such a method is robust, allowing the model to learn diverse and complex interactions without the dependence on a massive, fully annotated HOSI dataset, enabling strong generalization to unseen scenes.

**Weaknesses:**

Need to show some benefits on the downstream tasks such as robot learning, gaming, humanoid motion learning, e.t.c.

How is the model integrated with real physics?

And how is the model transfer to different simulation socially-inteactive environments (indoor vs outdoor, household, store, hospital, e.t.c.)?

**Questions:**

See weaknesses

---

> ### Author Response · Authors · 2025-11-21
> **Official Comment by Authors**
>
> Thank you for you thoughtful comments and questions.
>
> > Need to show some benefits on the downstream tasks such as robot learning, gaming, humanoid motion learning, e.t.c.
>
> Thanks for the helpful advice. We present some downstream applications below and have added a paragraph about the promising application at the end of Introduction Section in the revision.
>
> 1. Planner component for robot and humanoid motion learning.
> InfBaGel can act as a high-level planner to generate plausible human-like interaction motions. A downstream controller can then track or adapt these motions under its own dynamics and actuation constraints. In this view, InfBaGel provides a strong motion prior capturing rich human-object-scene interaction patterns, while the controller handles low-level physics and embodiment-specific details.
> 2. Control signals for video and animation generation.
> Because InfBaGel produces temporally consistent 3D motion, they can serve as a high-level control signal for video generation models. This stronger 3D guidance can enhance physical consistency for 2D video generation.
> 3. Character animation for films and games.
> The motion generated by InfBaGel provises an alternative to  expensive real human motion capture in film/game pipelines, especially for complex interactions with cluttered scenes, significantly reducing production costs.
>
> > How is the model integrated with real physics?
>
> Our model serves as a motion planner that can be coupled with physics-based controllers in real or simulated environments. Concretely, in a real-world robotic deployment, the integration can proceed as follows:
> 1. Scene acquisition and voxelization. A depth camera is used to capture the scene geometry, which is then voxelized into the same voxel representation as used during training.
> 2. HOSI motion generation with the proposed model. Given the voxelized scene, object geometry, and a language instruction, our model generates a human-object-scene interaction in kinematic form.
> 3. Retargeting to the robot embodiment. The generated human motion is retargeted to the robot or humanoid, producing a reference trajectory in the robot’s configuration.
> 4. Physics-based motion tracking via a controller. A low-level controller then tracks this reference trajectory under real dynamics, contact forces, and actuation constraints.
>
> > And how is the model transfer to different simulation socially-inteactive environments (indoor vs outdoor, household, store, hospital, e.t.c.)?
>
> The scene generalization capability of InfBaGel stems from its use of a voxelized scene representation and the hybrid data training strategy. Conditioning on this abstract representation, the model is decoupled from fine-grained scene geometry and fixed semantic categories. Consequently, it can be easily applied to  novel environments, like a household, store, or hospital, if the scene can be voxelized, without requiring additional architectural modifications or retraining.
>
> In Appendix A.1 (Figure 5), we have further added more visual results for socially-interactive scenes as suggested, including a store and a physical therapy room. These results and our HOSI benchmark with 67 unseen scenes validates our zero-shot scene generalization capability.

---

### Official Review · Reviewer_2xtj · 2025-10-31

**Soundness:** 3
**Presentation:** 3
**Contribution:** 3
**Rating:** 6
**Confidence:** 3

**Summary:**

The paper proposes a method to generate paired human-object interaction motion in the context of a static scene. The input is the static scene geometry, interaction object geometry, text instruction, and goal position of the object. The dynamic human-object-scene interaction is represented as a sequence of five voxel grids, representing the scene and object occupancy at the start position, goal position, and (optionally) 3 intermediate timesteps along the human-object interaction trajectory. Voxel grids at intermediate timesteps can be masked out for unconditional trajectry generation. Then, an auto-regressive consistency model is trained via consistency distillation to generate human and object motion conditioned on these inputs. To ensure strict collision avoidance, the denoised trajectories outputted by the consistency model are iteratively refined to avoid colliding with the static scene geometry, by introducing guidance from each colliding voxel to the nearest free-space. The method is trained on a combination of real-world HSI data (LINGO) and synthetic HOI data (OMOMO).

**Strengths:**

- Human-object-scene interaction generation is important and challenging problem.
- The paper achieves substantially improved results compared to LINGO and TRUMANS.
- The method is technically sound and the proposed components are effective: the dynamic perception encoder helps improve task success rate, the consistency model improves generation speed, and the bump-aware guidance reduces scene penetration.

**Weaknesses:**

- Despite the dynamic scene encoder and bump-aware guidance, the method does not seem to generate physically plausible manipulation motions. In Fig2(c) and Fig2(e), the crate is at a strange angle, hand poses do not seem to be predicted, and the hands are either penetrating with the crate or not in contact.
- The results are shown for a single task of moving an object from the start location to the goal location through a static scene. Therefore, it is not clear to me that the text instruction is necessary or useful in the problem formulation. More qualitative or quantitative examples for diverse interaction types (such as those seen in OMOMO - kicking and dragging objects, lifting over your head or in different manners) would strengthen the applicability of the method

**Questions:**

- The exact output format of the dynamic perception encoder has many missing details and is not fully clear in Sec. 3.2. It would be helpful to refer to the appendix in the main text and provide mathematical notations with the shape of each variable. My understanding is, each "voxel grid" is a 3D array {0,1,2}^(NxNxN) where N is the size of the voxel grid? And there are five voxel grids, corresponding to the start position, goal position, and three intermediate timestamps? How are the intermediate timestamps sampled?

---

> ### Author Response · Authors · 2025-11-21
> **Official Comment by Authors (1/2)**
>
> Thank you for your thoughtful comments and questions.
> > Despite the dynamic scene encoder and bump-aware guidance, the method does not seem to generate physically plausible manipulation motions. In Fig2(c) and Fig2(e), the crate is at a strange angle, hand poses do not seem to be predicted, and the hands are either penetrating with the crate or not in contact.
>
> We acknowledge that occasional non-contact and slight penetrations can be observed in a few cases. The main reasons are (i) current human-object interaction (HOI) dataset [1] lacks hand annotations, making explicit hand pose estimation and grasp modeling difficult; and (ii) motion capture may contain artifacts that introduce noise in motion data. Under these conditions, achieving perfect contact and zero penetration in a purely kinematic generation setting is challenging.
>
> Notably, compared with the kinematics‑based baselines [2, 3, 4, 5], our approach substantially improves physical feasibility, achieving the lowest human-object penetration and highest contact rates on both HOSI and HOI benchmarks (as shown in the following Table,  from Table 1 and Table 5 in the revised paper), demonstrating superior physical realism.
>
> | Method                                 |      C% ↑ | P_body ↓ |
> | -------------------------------------- | --------: | -------: |
> | *Comparision on the HOSI benchmark* |           |          |
> | TRUMANS [2]                                |     0.46 |     4.91 |
> | LINGO [3]                                 |     0.54|     5.72 |
> | Ours                               | **0.78** | **3.96** |
> | *Comparision on the OMOMO benchmark*|           |          |
> | LINGO [3]                                  |     0.41 |     4.80 |
> | CHOIS [4]                                 |     0.53 |     2.52 |
> | ROG [5]                                    |     0.54 |     5.86 |
> | Ours                                 | **0.60** | **2.49** |
>
> > The results are shown for a single task of moving an object from the start location to the goal location through a static scene. Therefore, it is not clear to me that the text instruction is necessary or useful in the problem formulation. More qualitative or quantitative examples for diverse interaction types (such as those seen in OMOMO - kicking and dragging objects, lifting over your head or in different manners) would strengthen the applicability of the method.
>
> Thank you for the suggestion. The HOSI benchmark (Table 1) is constructed from OMOMO-test and TRUMANS, and therefore inherits a wide range of interaction types, including lifting, kicking, dragging, rotating and so on, ensuring a comprehensive and representative evaluation. In the revision, we have clarified this point in Sec. 4.1 Evaluation Settings and Appendix A.5.
>
> Moreover, we have added additional visual comparisons in this revision: a dragging motion example in Figure 2 and a pushing motion example in Figure 3. Due to space constraints in the main paper, we include further examples, such as lifting object overhead, kicking, and other diverse interactions, in Figures 4 and 5 of Appendix A.1, across different scenes and object categories. These examples demonstrate the robustness and applicability of our method across varied instructions and scenes.
>
> Regarding the necessity of text instructions, they play an essential role in our formulation as high-level control signals specifying the intended interaction type. Without this condition, the model tends to collapse to the most frequent behaviors and loses controllability.
>
> *[1] Object Motion Guided Human Motion Synthesis. Li et al.*
>
> *[2] Scaling Up Dynamic Human-Scene Interaction Modeling. Jiang et al.*
>
> *[3] Autonomous Character-Scene Interaction Synthesis from Text Instruction. Jiang et al.*
>
> *[4] Controllable Human-Object Interaction Synthesis. Li et al.*
>
> *[5] Guiding Human-Object Interactions with Rich Geometry and Relations. Xue et al.*

---

> ### Author Response · Authors · 2025-11-21
> **Official Comment by Authors (2/2)**
>
> > The exact output format of the dynamic perception encoder has many missing details and is not fully clear in Sec. 3.2. It would be helpful to refer to the appendix in the main text and provide mathematical notations with the shape of each variable.
>
> As suggested, we have refined the presentation of the dynamic perception encoder with more details to clarify potential confusion in Sec. 3.2, and provided a reference to Appendix A.6. Specifically, we employ a ViT to independently encode these scene voxel grids to a 512-dimensional embedding. These embeddings are further concatenated to a 5x512 dimensional representation of scene.
>
> > My understanding is, each "voxel grid" is a 3D array {0,1,2}^(NxNxN) where N is the size of the voxel grid?
>
> Yes. Concretely, each scene voxel grid is a 3D array $\mathcal{S} \in \{0, 1, 2\}^{N \times N \times N}\$, where $ N = 32 $ is the grid size. We use this $ 32 \times 32 \times 32 $ grid to represent a cubic region of side length $1.2$ m centered at the pelvis or goal position, so each element corresponds to a cube of side length $ 1.2 \div 32 = 0.0375 $ m, indicating whether that voxel is occupied or free.
>
> > And there are five voxel grids, corresponding to the start position, goal position, and three intermediate timestamps? How are the intermediate timestamps sampled?
>
> Yes. Within each generation window, we use five scene voxel grids to represent local scene information, three dynamic and two static. Specifically, the two static voxel grids respectively capture the start and goal regions, while the three dynamic voxels are centered at the pelvis positions of intermediate frames uniformly sampled within the temporal window.  Concretely, for a window of $T$ frames, we take the scene voxel grids centered at the pelvis position of frames $T/3$, $2T/3$, and $T$ (where $T$ denotes the end of the current window, not the end of the entire sequence). The dynamic voxels are masked out during coarse prediction, and are updated iteratively as the prediction is refined. We have revised it in Sec. 3.2 and Appendix A.6  to make it clear.

---

### Author Response · Authors · 2025-11-21
**General Response by Authors**

We sincerely thank all reviewers for their insightful comments and positive recognition of our work, and we also express our gratitude to the Area Chairs and Program Chairs for their time and dedication.
We are encouraged that reviewers recognize InfBaGel across three key aspects:
- **Motivation:**
  - `R2xtj` points out that human-object-scene interaction (HOSI) generation is an important and challenging problem.
  - `R73R8` appreciates that our motivation is clear: to generate higher quality HOSI and to tackle data scarcity of this field.
  - `Rf1br` recognizes our motivation as well-presented.
- **Method:**
  - `R9B2s` acknowledges our novel and robust approach that integrates dynamic perception with coarse-to-fine iterative refinement, yielding more physically consistent and realistic HOSI with strong generalization to unseen scenes, without the dependence on a massive, fully annotated HOSI dataset.
  - `R2xtj` appreciates that our method is technically sound and the proposed components are effective: the dynamic perception encoder helps improve task success rate, the consistency model improves generation speed, and the bump-aware guidance reduces scene penetration.
  - `Rf1br` recognizes the technical method as correct and `R73R8` notes that the contribution is clear.
- **Results:**
  - `R2xtj` acknowledges that our work achieves substantially improved results compared to LINGO and TRUMANS.
  - `R9B2s` highlights our work leading to more physically consistent and realistic HOSI and strong generalization to unseen scenes.
  - `R73R8` points out that our work is significantly better than previous works in terms of collision avoidance.

---
We have revised the paper according to feedback (changes are highlighted in blue) with the key revisions summarized as follows:

**1. Presentation Refinement**
- To address `R73R8`'s question on "confusion of motion consistency model",  we have revised Section 3.3 to improve clarity. The updated subsection now clearly explains the coarse-to-fine generation process of the diffusion model with dynamic perception, the motivation for consistency distillation to enhance inference speed and scene perception, and the corresponding objective. Additionally, we renamed Fig. 1(a) to Auto-regressive Motion Model to better align with the text and avoid confusion.
- To address `R2xtj`'s question on "exact output format of the dynamic perception encoder", we have added more details and notations in Sec. 3.2 and Appendix A.6 to clarify the input voxel grid format and output embedding shape. We have also clarified how the intermediate voxels sampled, masked in coarse prediction, and iteratively updated during refinement.

**2. Experimental Setting Clarification**
- To address `R2xtj`'s question on "interaction diversity", we have clarified in Sec. 4.1 and Appendix A.5 that the HOSI benchmark is built from OMOMO-test and TRUMANS, covering all interaction types in OMOMO (e.g., kicking, dragging, lifting, etc.), supporting comprehensive evaluation beyond the single "move from start to goal" task. Given this multi-interaction setting, text instructions serve as high-level semantic conditions that specify the intended interaction type for controllable behavior generation.

**3. More Qualitative Evaluations**
- To address `R2xtj` and `R73R8`'s questions on "more qualitative results", we have added diverse visual examples across scenes, motion, and object types to demonstrate the method's robustness and generalization capabilities. We have further added dragging and pushing in Figs. 2-3 in the revised paper and additional interactions in different scenes, including lifting overhead, kicking, etc., in Figs. 4-5 in Appendix A.1.
- To address `R9B2s`'s question on "transfering to socially-interactive scenes", we have added visual results in a store and a physical therapy room in Fig. 5 in Appendix A.1.

**4. Reorganized Ablation Study and Analysis of "C+B"**
- To address `Rf1br`'s question on "each contribution's effectiveness", we have reorganized the ablation study in Sec.4.3 to systematically evaluate the effectiveness of each component (dynamic perception encoding, bump-aware guidance and consistency model) with the results in Tab. 3 and Tab. 4. We also have added a visual comparison of ablationed variants in Fig. 3.
- To address `Rf1br`'s question on "C+B results", we have added a dedicated paragraph in Sec. 4.3 (L. 514-518) explaining the trade-off between goal accuracy and physical plausibility. We clarified that this balance is desirable for downstream applications (e.g., robotics and animation), where collision alleviation is often more critical than marginal goal distance gains.

**5. Discussion on Applications**
- To address `R9B2s`'s question on "downstream applications", we have added a discussion on downstream tasks at the end of the Introduction, outlining how our model serves as a high-level planner for robotics and discussing its adaptability to diverse daily life scenarios.

---

### Comment · Area_Chair_pgbY · 2025-11-27

Dear Reviewers,

Thank you for your thoughtful evaluations of this submission. The authors have provided their responses and clarifications during the discussion phase. To ensure a well-informed final decision, I kindly encourage you to continue the discussion by reviewing the authors’ replies and adding any follow-up thoughts you may have.

If any of your original concerns remain unresolved, please feel free to raise them directly in the discussion thread.

Thank you again for your time and valuable contributions.

Best regards,
Area Chair

---

### Meta-Review · Area_Chair_iiM4 · 2026-01-07

**Summary:**

The paper got positive ratings from all reviewers (6,6,6,6).

Reviewers acknowledged the well-justified motivation of the paper, the methodological soundness of the proposed approach, and its superior performance compared to baseline methods.

The major concerns raised by the reviewers include: (1) a lack of clarity in certain methodological details and the evaluation setup; and (2) insufficient qualitative results to fully demonstrate the practical effectiveness of the proposed method.

**Reviewer Concerns:**

The AC found that the authors’ rebuttal mostly resolved the major concerns raised by the reviewers. The authors’ responses and the revised manuscript provide additional details on the method and experimental settings, as well as further ablation studies and qualitative results.

**Reviewer Scores:**

Overall, the AC believes that the authors’ responses are convincing and clear. Given that all reviewers already gave positive ratings, the ratings are expected to remain unchanged.

---

### Decision · Program_Chairs · 2026-01-26

Accept (Poster)